# Bacteriophage-Mediated Cancer Gene Therapy

**DOI:** 10.3390/ijms232214245

**Published:** 2022-11-17

**Authors:** Gleb Petrov, Maya Dymova, Vladimir Richter

**Affiliations:** The Institute of Chemical Biology and Fundamental Medicine, Siberian Branch of the Russian Academy of Sciences, 630090 Novosibirsk, Russia

**Keywords:** bacteriophages, cancer gene therapy, gene delivery

## Abstract

Bacteriophages have long been considered only as infectious agents that affect bacterial hosts. However, recent studies provide compelling evidence that these viruses are able to successfully interact with eukaryotic cells at the levels of the binding, entry and expression of their own genes. Currently, bacteriophages are widely used in various areas of biotechnology and medicine, but the most intriguing of them is cancer therapy. There are increasing studies confirming the efficacy and safety of using phage-based vectors as a systemic delivery vehicle of therapeutic genes and drugs in cancer therapy. Engineered bacteriophages, as well as eukaryotic viruses, demonstrate a much greater efficiency of transgene delivery and expression in cancer cells compared to non-viral gene transfer methods. At the same time, phage-based vectors, in contrast to eukaryotic viruses-based vectors, have no natural tropism to mammalian cells and, as a result, provide more selective delivery of therapeutic cargos to target cells. Moreover, numerous data indicate the presence of more complex molecular mechanisms of interaction between bacteriophages and eukaryotic cells, the further study of which is necessary both for the development of gene therapy methods and for understanding the cancer nature. In this review, we summarize the key results of research into aspects of phage–eukaryotic cell interaction and, in particular, the use of phage-based vectors for highly selective and effective systemic cancer gene therapy.

## 1. Introduction

Cancer is currently one of the leading causes of mortality in people worldwide. According to studies, 19.3 million new cases and 10 million deaths from cancer were registered in the world in 2020 [1]. The population aging that is observed throughout the world and the growth in the number of carcinogenic factors of a biological and chemical nature in the environment induce the growing trend of the annual increase in oncological diseases. Despite the increase in the effectiveness of method for detecting malignant neoplasms in the early stages and the advanced study of the cancer pathogenesis, a wide variety of tumor diseases and their causative factors prevent the invention of a single and comprehensive therapeutic approach.

Gene therapy is an experimental personalized approach to cancer treatment with the potential to provide directed tumor destruction [2]. The invention of targeted vectors for the delivery of drugs, functional peptides and transgenes can provide an effective and, most importantly, more selective and safer cancer therapy than traditional chemotherapy. To date, the most popular vehicles are eukaryotic viruses [3]. The local intratumoral administration of natural infectious agents of mammalian cells facilitates the efficient delivery and expression of therapeutic genes into cancer cells. However, the systemic administration of these vectors is accompanied by a decrease in the selectivity of delivery, in connection with the wide tropism of eukaryotic viruses to the cells and tissues of host organisms, the absorption of their particles by the liver and the reticuloendothelial system of the body and the triggering of a powerful immune response, which complicates repeated injections [4,5].

Various studies are an indication that bacteriophages can be used to create vectors that provide an efficient targeted delivery of genes and drugs to cancer cells following systemic administration [6]. The structure and biology features of phages and the proven safety of their use in various medical applications distinguish them as a potentially powerful tool for cancer therapy. Bacteriophages are actively used to develop cancer vaccines [7]. The extremely high stability and resistance of phages to environmental conditions is used to create virus-like particles (VLPs) that selectively deliver various therapeutic cargoes to tumors. Capsids of phages such as MS2 and P22 have been successfully utilized to deliver therapeutic non-coding RNAs to HCC cells and cytochrome P450 to cervical carcinoma cells, respectively [8,9,10]. Bacteriophages have found application in photodynamic cancer therapy based on the formation of reactive oxygen species in tumor tissue as a result of the interaction of light and a specific photosensitizer [11]. The selective phage-mediated delivery of these photosensitizers to breast cancer cells provided a noticeable decrease in tumor viability. There are many phage-mediated approaches to selective cancer therapy; however, the most interesting are methods based on the interaction of the virus genetic material with cellular molecular machinery. In this review, we focused on highlighting the latest data on the use of phages as vectors for cancer gene therapy, the action of which is mediated by the targeted expression of engineered prokaryotic viruses’ transgenes in tumor cells. In addition, we describe the mechanisms of the entry and expression of bacteriophages’ genetic material in eukaryotic cells, underlying the therapeutic applications of these viruses.

## 2. Bacteriophages: Structure and Biology

Bacteriophages are obligate parasites of bacteria and represent the most numerous group of biological entities inhabiting the biosphere [12]. Since their genomes completely lack the genes necessary for independent reproduction, phages use the molecular machines of host cells to express their own genes. Bacteriophages are extremely diverse and infect almost all existing bacteria [13]. These viruses play a huge role in the evolution of prokaryotic organisms, influence the interaction of bacteria with each other and are able to impact the interaction of bacteria with multicellular organisms [14]. The unique properties of bacteriophages have made them a promising tool in various fields of biotechnology [15]. Moreover, phages and anti-phage defense systems of bacteria have had a significant impact on the emergence and development of genetic engineering [16].

A key role in the popularity of bacteriophages was played by the peculiarities of their structure and the organization of their genomes. From a morphological point of view, bacteriophage particles have a well-defined three-dimensional structure (Figure 1).

The vast majority of virions consist of a head and a tail [17]. The head contains the genetic material of the virus, surrounded by a capsid. The genetic material may be single- or double-stranded DNA or RNA [12]. An important element of the bacteriophage structure is the baseplate located at the end of the tail, from which thin long fibrils extend, which facilitate the attachment of the phage to the bacterium [18]. The baseplate itself contains receptor-binding proteins responsible for recognizing specific molecules on the surface of the bacterial membrane. The interaction of bacteriophages with bacterial cells is extremely specific: the victim must express the corresponding receptor on the surface of its membrane, which ensures the binding and internalization of the phage [19]. There are also bacteriophages that do not have a tail. These viruses display receptor-binding proteins directly on the surface of their capsids [20]. The interaction of a phage with a host bacterium occurs in several stages [21]. After the primary contact, mediated by the drift of phage particles in the solution and Brownian motion, electrostatic forces cause reversible and nonspecific binding of the phage to the bacterium [22]. This interaction later becomes irreversible due to the binding of the elements of the virus capsid to the surface receptors of the cell, which, depending on the type of phage, can be represented by: amino acids, teichoic acids, glycoproteins, lipopolysaccharides or pili fragments [23].

The baseplate coordinates not only the recognition of specific receptors on the surface of the cell membrane but also the binding of the phage particle to the host bacterium. Successful binding to the bacterial receptor triggers a conformational change in the baseplate that eventually results in tail shortening. The signal for the deployment of these events is a change in the orientation of the fibrils with respect to the baseplate [22]. It should be noted that this fibril reorientation occurs exclusively on the surface of the host cell and is not observed in the free state of viral particles in the solution. Moreover, no chemical energy is used to change the fibril orientation and baseplate conformation. There are several proposed mechanisms that explain these particularities of the phage binding to the host cell. According to one, the strong binding of a phage to a bacterium is achieved by the partial binding of virus fibrils to specific receptors on the cell surface, while the orientation of the entire virion on the cell surface and the further interaction of fibrils with cell receptors occur under the influence of the movement of the medium. In the context of this mechanism, the restriction of the phage particle movement contributes to a change in the conformation of unbound fibrils toward their greater affinity for the cell surface [22]. According to another proposed mechanism, the change in the three-dimensional conformation of the baseplate and the reorientation of the virus fibrils towards the cell, leading to the strong binding of the phage to the bacterium, occur under the influence of divalent cations, especially Ca^2+^. The fact is that some surface polysaccharides and proteins of bacterial cells bind divalent cations, significantly increasing their concentration in the area of interaction between the phage and the host. It has been established that, for some bacteriophages, the presence of calcium ions is a necessary condition for the infection of host cells [24]. The efficiency of interaction between bacteriophages and prokaryotic cells is also influenced by other ions present in the solution: the higher the ionic strength of the environment, the more infectious the phage particles become [25]. It is worth noting that these mechanisms are not mutually exclusive and can coexist together.

After the irreversible binding of the phage to the bacterium, the contraction of the protein coat of the virus tail occurs, during which the protruding rigid tail tube pierces the outer membrane of the cell. Next, the tube penetrates the periplasmic space and, with the help of enzymes located at its end (mainly lysozyme), locally destroys the peptidoglycan of the cell wall. The genetic information of the virus is then injected into the cytoplasm of the bacterium, while the protein shell of the phage remains outside [26].

Tailless phages infect host cells in a different way. Filamentous bacteriophages (Ff-phages), for example, use cell pili to bind to and enter bacterial hosts. Ff-phages are known to bind *E. coli* F-pilus with their minor pIII protein. The binding of the virus to the bacterial pilus likely induces its retraction. In this case, the end of the filamentous phage virion, which exposes the pIII protein, is in the periplasmic space of the cell. In this layer, the phage binds to the secondary receptor, the TolA protein, which is part of the TolQRA complex, anchored in the inner cell membrane. Presumably, the pIII protein is also responsible for the formation of pores in the inner cell membrane, which are necessary for the further injection of the genetic material of the virus into the cell [27].

Regardless of the mechanism of phage genetic information penetration into the host cell, there are two possible methods of its further implementation [28]. When the lytic cycle is initiated, a complete restructuring of the cell metabolism occurs; all the energy of the cell is directed to the replication of the genetic material of the virus, the transcription of viral genes and the translation of viral proteins. The final stage of this cycle is the assembly of mature virions and their release by cell lysis. The lysogenic cycle is characterized by the reversible interaction of the phage genome with the bacterial genetic system. In this case, the viral genome replicates synchronously with the host’s DNA, while the integrity of the cell is maintained. The lysogenic cycle in the vast majority of cases continues until prophage induction and switching to the lytic pathway occur.

A detailed study of the bacteriophages structure, their interaction with bacterial hosts and their life cycle makes it possible to actively use these viruses in many areas of modern applied and basic research. Due to the growing antibiotics resistance of bacteria, phage therapy for infectious diseases is currently experiencing a renaissance [29]. The effectiveness of the use of bacteriophages in the diagnosis of food safety has been proven [30]. Phages find their application in plant pathogen control, biomedical diagnostics, molecular biology and genetic engineering [31,32,33].

Currently, bacteriophages are considered as promising vectors for cancer gene therapy [5]. They are believed to have a number of advantages over non-viral and eukaryotic virus-based vectors. First of all, phage-based vectors compare favorably with eukaryotic viruses because of the absence of natural tropism for eukaryotic cells, which increases the selectivity of therapeutic cargo delivery to target cancer cells. Although it was previously thought that phages could not interact with organisms that are more complex than prokaryotes, in recent years, there has been increasing evidence that natural bacteriophages can directly interact with cells of higher organisms [34].

## 3. Bacteriophage–Eukaryotic Cells Interactions

The first evidence of the ability of phages to directly interact with mammalian cells dates back to the 1940s and was provided by Bloch [35]. This study discovered the ability of bacteriophages to accumulate in cancerous tissues and inhibit tumor growth. Decades later, Kantoch established the ability of phages to bind guinea pig leukocytes and internalize into them. More recent studies have confirmed the high frequency of phage interactions with cells of the mammalian immune system [36]. In the 1970s, C. R. Merril demonstrated the ability of the lambda phage to interact with human fibroblast cells [37,38]. For a long time, however, the molecular mechanisms of interaction between bacteriophages and eukaryotic cells remained unexplored. Phages are able to infect only those bacteria that have virus-specific receptors on the surface of the cell membrane. Since eukaryotic cells have a number of structural features that fundamentally distinguish them from bacterial cells (in particular, unique cell surface markers), it was not clear how bacteriophages could interact with them. However, the accumulated evidence to date suggests that bacteriophages can specifically bind to eukaryotic cells.

### 3.1. Binding and Entry of Phage into Eukaryotic Cells

The possibility of the specific binding and penetration of the engineered bacteriophage into mammalian cells was first shown in a study conducted by D. Larocca et al. [39]. However, the first evidence for natural bacteriophages having mechanisms for specific binding and penetration into eukaryotic cells appeared only a few years later. In 2003, Gorski and colleagues formulated a hypothesis according to which some bacteriophages (in particular, the T4 phage) express a KGD peptide (Lys-Arg-Gly) with an affinity for the β3 integrin and are able to bind eukaryotic cells that carry this cellular marker with high specificity [40]. Moreover, since β3 integrin is very common in cancer cells and is considered as one of the possible factors contributing to their metastasis, specific phage binding to this protein, according to this hypothesis, should lead to the inhibition of the tumor cell spread. This hypothesis was soon experimentally confirmed when the ability of the bacteriophages T4 and HAP1 (a substrain of T4 phage) to bind mouse and human melanoma and lung cancer cells and prevent their metastasis was discovered. This specific binding is mediated by the interaction of the GP24 capsid protein, which contains the KGD motif, with β3 integrin, which is highly expressed on the surface of tumor cells [41].

Integrin β3 is not the only surface protein of eukaryotic cells that promotes their specific interaction with bacteriophages. It is known that T4 and M13 phages can suppress the expression of the HSP90 gene in human prostate cancer (PC3) cells, which is responsible for the stimulation of mitosis, DNA repair and the prevention of apoptotic events. Such suppression is due to the specific binding of these phages to the HSP90 receptors [42].

Another bacteriophage, PK1A2, has been shown to be able not only to bind specifically but also to penetrate into kSK-N-SH human neuroblastoma cells [43]. These cells exhibit large amounts of polysialic acid on the membrane surface. This homopolymer, which is a particular post-translational modification of the neural cell adhesion molecule, has a structural similarity to *E. coli* K1 lipopolysaccharide, providing the specific binding of the phage to the host cell. The binding of PK1A2 to polysialic acid initiates endocytosis, as a result of which the bacteriophage successfully enters the cell. Phage binding to β3 integrin and the HSP90 receptor should also initiate receptor-mediated endocytosis events leading to the entry of phage particles into the eukaryotic cell, but this assumption has not yet been experimentally confirmed.

Bacteriophages can also use the mechanisms of nonspecific penetration into mammalian cells. The ability of some bacteriophages to freely penetrate into eukaryotic cells and overcome entire cell layers in the process of transcytosis is known [44]. These viruses include, in particular, T4, T5, T7, SP01, SPP1 and P22 phages. The bacteriophage absorbed by the vesicle moves in the cytoplasm from one pole of the cell and is released at the other. Transcytosis is a dose-dependent process and proceeds mainly in the apical-basolateral direction, as was shown for bacteriophages in studies on monolayer epithelial cells of various human organs [44].

In addition to transcytosis, phages also use other methods of nonspecific penetration into eukaryotic cells. The M13 filamentous phage is characterized by strong plasticity in choosing the mechanism of internalization into mammalian cells. It is known that it can enter both epithelial and endothelial cells by clathrin-mediated endocytosis, macropinocytosis or caveolae-mediated endocytosis (in the case of endothelial cells) [45]. Another filamentous phage, Pf4, is able to internalize into monocytes by clathrin-mediated endocytosis [46]. It should be noted that, regardless of the mechanism of penetration, the rate of this process is apparently influenced by both the type of eukaryotic cells and the size of phage particles [47]. At present, bacteriophages have been found in endosomes, lysosomes, the Golgi apparatus, the cytoplasm and the nucleus of various mammalian tissue cells [48].

### 3.2. Phage Intracellular Activity

Inside the eukaryotic cell, bacteriophages encounter unfavorable environmental factors that seriously limit their activity in the cytoplasmic space. Like some eukaryotic viruses [49], phage particles penetrating the cell become isolated in endosomes and phagosomes, where they undergo further degradation. Since bacteriophages are stable under the harsh conditions of the acidic pH characteristic of late endosomes and phagolysosomes, the activity of lysosomal proteases is presumably the main factor in the destruction of viral particles [48].

Although most bacteriophages are degraded after internalization into mammalian cells, there are data indicating a close interaction between phages and eukaryotic intracellular proteins. For some phages, their ability to activate Toll-like receptors localized on the endosome membrane and induce a cellular immune response has been proven [46,50]. At the same time, the genetic material of some phages or even entire viral particles can leave the endosome and be released into the cytoplasmic space of the eukaryotic cell [48]. While eukaryotic viruses have well-studied evolutionary mechanisms for the release of genetic material, whether it is the fusion of the viral membrane with the host endosomal membrane in the case of enveloped viruses, or lysis and permeabilization of the endosomal membrane in the case of non-enveloped viruses, it is still unknown what factors ensure the endosomal escape of phages [51]. Solving this problem is an important issue of phage genetic engineering, which is aimed at creating highly efficient vehicles for the delivery of therapeutic cargoes into mammalian cells.

It is assumed that the genetic material of bacteriophages can interact with cytosolic proteins, participating in the activation of cGAS-STING and RIG-I pathways [52]. However, it is still unclear how the undressing of bacteriophages occurs in eukaryotic cells. Apparently, endosomal proteases and cytosolic proteasomes may be involved in this process [48]. It has been established, for example, that cathepsin L is capable of destroying bacteriophage capsids in vitro [53]. On the other hand, it has been repeatedly shown that the use of protease and proteasome inhibitors can enhance phage-mediated gene expression [53,54,55]. Obviously, the solution of this problem requires more detailed research in the future.

Another important and little-studied aspect of phage–eukaryotic cell interaction is the transfer of bacteriophage genetic material to the cell nucleus. The phage genome, like the genomes of eukaryotic viruses, can enter the nucleus during mitosis, while the nuclear envelope is temporarily destroyed [56]. Another possible mechanism is based on the specific interaction of phage terminal proteins (TPs) with eukaryotic transport factors. TPs are covalently linked to the genomes of some phages and play an important role in the packaging and replication of the genetic material of these viruses. The detection of nuclear localization signals (NLSs) in the bacteriophages’ TPs and the demonstration of their full functionality in eukaryotic cells prove the existence of a mechanism for the directed nuclear translocation of phage genomes [57]. In addition, TPs bound to phage DNA appear to protect the genetic material of the virus from exonucleolytic degradation in the cytoplasm of the eukaryotic cell. This discovery has implications for therapeutic phage engineering as well, since the creation of TP-like DNA-associated proteins in the bacteriophage structure and the insertion of NLSs in their amino acid sequences may contribute to the improved delivery of transgenes by these viruses.

It is now becoming clear that the interaction between bacteriophages and eukaryotes is much more complex and diverse than previously thought. The presence of functional NLS in the proteins of some phages and the discovered ability of phage genomes to integrate into mammalian DNA indicate the close interaction of prokaryotic viruses with eukaryotes and the existence of an active direct horizontal gene transfer between them [58]. It has been established that bacteriophage genes can not only penetrate into the nuclei of eukaryotic cells but also be expressed in them [38,59]. Unfortunately, to date, the mechanisms underlying phage DNA transcription in mammalian cells have been studied very poorly. It is assumed that some phage genomes may contain sequences homology to eukaryotic transcription factor binding sites that mediate the interaction of RNA polymerase II with viral DNA. An analysis of the prophage sequence found in the genome of enterohemorrhagic Shiga toxin-2 (Stx2) producing *E. coli* revealed the presence of TATA-box, eukaryotic transcription factor sites and polyadenylation recruitment sequences [60]. This Stx2 promoter was found to be fully functional in eukaryotic cells. It was shown that the expression of the reporter protein and Stx2 genes under the control of the bacteriophage promoter led to the corresponding functional proteins synthesis in transfected mammalian cells. Future work aimed at achieving a better understanding of the nature of the phage–eukaryotic cells interaction will have to focus on the identification and study of the prokaryotic viruses’ gene expression molecular mechanisms in the cells of higher organisms. Another equally intriguing phenomenon is the specific interaction of bacteriophages with cancer cells. Recent studies have found that the interaction of M13 and T4 phages with LNCaP (Lymph Node Carcinoma of the Prostate) cancer cells changes the expression level of a number of cellular factors in these cells and influences their survival [61]. The further study of these phage–eukaryotic interaction aspects is necessary for the interpretation of previously poorly explained states of cancer patients. Moreover, a deep study of the bacteriophages–tumor cells interaction mechanisms should contribute to the development of existing approaches and the creation of new approaches to phage-mediated cancer gene therapy.

## 4. Bacteriophage-Mediated Cancer Gene Therapy

### 4.1. Phage Display

Gene therapy is an experimental approach to treating various diseases by introducing foreign therapeutic genes or manipulating disease-related genes [2]. The most attractive field of gene therapy application today is the treatment of cancer [62]. Bacteriophages have played a significant role in the formation and development of various cancer gene therapy approaches. One of the most important factors in the development of this medical oncology area is the creation of phage display technology in 1985 [63]. Currently, this technology is widely used to study protein–protein, protein–peptide and DNA–protein interactions, as well as to create a new generation of diagnostic and immunotherapeutic drugs [64]. This technology is based on the insertion of foreign nucleotide sequences into one of the genes encoding bacteriophage capsid proteins. This produces a heterogeneous mixture of phage particles, each of which exposes its own peptide on the surface, encoded by the integrated DNA fragment. The phage display libraries thus obtained are screened, leading to the identification of peptides and their molecular targets. Immobilized targets, which can be represented by proteins, cells and tissues, allow for the affinity selection of phage-exposed peptides, during which non-immobilized viruses are washed away and fixed viruses enter a new cycle of affinity selection [65]. The invention of the phage display became possible due to several features of filamentous Ff phages: (1) the reproduction of these viruses is carried out without lysis of the host cell, which makes it possible to produce high titers of the phage in the laboratory [66]; (2) phage capsid proteins are amenable to genetic fusion with other peptide sequences, resulting in the exposure of the foreign protein on the surface of the virus; (3) the precisely established phenotype–genotype link makes it possible to select a specific phage from huge libraries and determine the primary sequence of the studied peptide and its encoding sequence [67]. Currently, for phage display, in addition to the filamentous M13 phage, bacteriophages such as T4, T7 and lambda phage are used [68].

Currently, phage display technology is widely used to search for functional antitumor peptides, surface markers of cancer cells and their highly specific ligands. Random peptide libraries presented on T7 phage, for example, contributed to the discovery of a potent selective inhibitor of tumor K-RAS, which is one of the main growth factors in various types of cancer [69]. At the same time, bacteriophages can be utilized not only as a tool for detecting functional peptides but also as their selective carriers. The display of specific peptides and proteins by phage capsids finds its application in the development of inhibitors of cancer-related signaling pathways. In a recent study, T4 phage was used as an inhibitor of one of the most important signaling pathways for tumor neovascularization, VEGF/VEGFR2 [70]. The developed recombinant bacteriophage displayed the extracellular domain of VEGFR2, thus providing the competitive inhibition and capture of VEGF, a key factor in tumor angiogenesis overexpressed by cancer cells. The production of immunogenic phage particles that exhibit foreign antigens on the surface of their capsid formed the basis for the creation of phage display-vaccines [7]. The systemic administration of such modified phage particles is capable of inducing a potent immune response through uptake and processing by antigen-processing cells (APCs), followed by the presentation of the resulting peptides through the major histocompatibility complex (MHC) class I and II pathways [71]. This principle underlies the work of cancer vaccines obtained by incorporating the tumor-associated antigen into the capsid structure of the bacteriophage. MHC-mediated peptide presentation stimulates both cellular and humoral immune responses. This principle underlies the work of anticancer vaccines derived by incorporating the tumor-associated antigen into the capsid structure of the bacteriophage. Currently, phage display vaccines have been successfully used in the immunotherapy of various types of cancer—in particular, hard-to-treat ones such as HER2-positive breast cancer and hepatocellular carcinoma [72,73]. The main advantages of such phage-based vaccines that increase their effectiveness on the immune system are the intrinsic adjuvant properties of the filamentous phage capsid proteins and the ability to display multiple copies of the antigen on the viral surface. There is another approach to creating anticancer vaccines based on bacteriophages, carried out by cloning the coding sequence of the antigen under the control of a strong eukaryotic promoter into the viral genome [71]. The phage DNA vaccines thus obtained are capable of inducing a potent anticancer immune response by expressing a foreign cancer-associated antigen within the APC.

### 4.2. Phage-Based Vectors for Cancer Gene Therapy

The development of phage-based cancer vaccines appears to be one of the most significant and promising areas of modern biotechnology and medicine [7]. At the same time, the phage display technology and bacteriophages themselves have also found their application in the creation of high-precision vectors of cancer gene therapy based on the transfer of therapeutic genes and their expression in cancer cells (Table 1).

Although the use of high-precision cancer-specific nanocarriers as delivery vehicles for therapeutic genes is a promising area of cancer therapy, clinical trials of gene therapy using in vivo gene delivery revealed an extremely low level of tumor targets transduction [83]. Since the effectiveness of gene therapy largely depends on the effectiveness of the therapeutic transgene carrier, the development of delivery methods is the most important issue of this approach. The most promising in this respect were viral vectors [3]. Eukaryotic viruses, as infectious agents of higher organisms, have developed complex and numerous mechanisms of interaction with cellular structures in the course of evolution. Compared to non-viral gene delivery methods such as the injection of naked DNA, gene gun and chemical methods, eukaryotic viruses show increased transfection efficiency in eukaryotic cells and the stable expression of therapeutic genes [84]. Lentivirus, adenovirus and AAV are the most preferred for cancer gene therapy because they have unique properties as vectors for delivering therapeutic genes to mammalian cells [3]. However, the targeted delivery of animal/human viruses has been hampered by their natural tropism for eukaryotic cells [4]. In addition, the use of vectors based on these viruses is limited by the immunogenicity of their systemic administration and the absorption by unwanted tissues and organs, such as the liver [6]. Other serious problems associated with the utilization of therapeutic drugs based on eukaryotic viruses include: oncogenicity, extremely limited capacity of the viral genome for transgene cloning, weak resistance of the capsid structure during its modification and the need for helper viruses [5].

Despite the fact that bacteriophages were considered poor vectors for transducing cells of higher organisms, they unexpectedly suggested a new way to develop methods for delivering therapeutic genes. The lack of tropism for mammalian cells, the higher cloning capacity, the easy modifiability and the phage display technology distinguish bacteriophages as a promising cancer gene therapy tool [6]. The production of phage particles carried out in bacterial cells is characterized by ease, speed and low cost. The systemic administration of bacteriophages is safe and has been successfully used in phage therapy for infectious diseases [29]. For a long time, however, it was not clear how to take advantage of these viruses, since phage-based gene therapy vectors showed extremely low levels of transgene expression in target cancer cells [39,85,86]. The solution came in 2006 when Hajitou and colleagues hypothesized that fusing the genomes of prokaryotic and eukaryotic viruses could lead to the creation of chimeric virus particles that combine the strengths of both types of original vectors [74]. The genome of this hybrid virus was obtained by inserting a recombinant AAV transgenic cassette containing the herpes simplex virus thymidine kinase (HSVtk) gene under the control of the CMV promoter and flanked by two full-length inverted terminal repeats (ITRs) from AAV serotype 2 into the M13 filamentous phage DNA. The specificity of transgene delivery was mediated by the RGD4C peptide exposed by the pIII protein of the phage capsid. This peptide targets the αvβ3-integrin receptor, overexpressed by tumor cells and cells of their vascular environment, but not by healthy tissue cells [87]. The result of the work was an extremely effective stable tumor-specific expression of the HSVtk gene, which simultaneously performs two functions: (1) a reporter gene for molecular PET imaging and (2) a tumor cell suicide gene in combination with ganciclovir (GCV). This approach provided the effective suppression of implanted tumors in laboratory mice and rats [74]. Later, this chimeric virus demonstrated its versatility as a therapeutic vector and found application in various cancer gene therapy strategies (Figure 2).

The efficiency of the AAV/phage (AAVP) vector in suicide cancer gene therapy has been repeatedly confirmed in subsequent studies. AAVP-RGD4C-HSVtk, in combination with GCV, had a pronounced antitumor effect in preclinical tests on mouse models of Kaposi’s sarcoma, bladder and prostate carcinoma and breast tumors [74], nude rats bearing human sarcoma xenografts [88], rat glioblastoma cells, mouse models of human glioblastoma and human melanoma cells [54,89]. Targeted HSVtk/GCV therapy has long been recognized as an effective gene therapy approach in the treatment of cancer [90]. The mechanism of the antitumor activity of HSVtk is its ability to convert GCV to GCV-triphosphate in cells expressing this enzyme. This cytotoxic metabolite inhibits DNA synthesis, which subsequently leads to cellular apoptosis. Moreover, cytotoxic GCV-triphosphate is able to spread from transduced cells to neighboring non-transduced cells via cell gap junctions or apoptotic vesicles, making this type of cancer gene therapy even more effective. This phenomenon, known as the “bystander effect”, was taken into account and used when targeting AAVP to mouse mammary tumor endothelial cells [75]. Since solid tumors are characterized by active angiogenesis, tumor-associated endothelial cells, which also overexpress surface integrin receptors, are an attractive target for cancer gene therapy. AAVP proved to be an effective and safe vector for delivering the HSVtk transgene to endothelial cells surrounding isogenic EF43-FGF4 mouse mammary tumors [75]. The subsequent treatment of this tumor model with GCV resulted in almost 80% cancer and endothelial cell death. EF43-FGF4 cells themselves have a low expression level of integrin receptors, but due to the bystander effect, they are sensitive to suicide gene therapy mediated by AAVP-RGD4C-HSVtk. As a result of this work, it was concluded that the effectiveness of cancer suicide gene therapy using AAVP is not limited by the efficiency of tumor cells transduction itself and increases due to the transduction of the tumor microenvironment and the heterotypic “bystander effect”.

Even greater success in AAVP-mediated suicide gene therapy has been achieved with a combination of ligand-directed tropism and transcriptional targeting [89]. The use of tumor-specific promoters in cell-targeting vectors has been noted as a promising strategy in cancer gene therapy [91]. One of the most attractive tumor-specific promoters is the Grp78 promoter, induced by stress and highly activating in a large number of tumor cells [92]. The use of this promoter in the RGD4C-AAVP vector provided a higher and more stable expression of HSVtk compared to the CMV promoter [89]. The expression of the transgene under the control of Grp78 did not weaken over time, which cannot be said about the use of viral promoters. At the same time, the Grp78 promoter increased the effectiveness of suicide HSVtk/GCV cancer therapy not only due to the increased expression of the therapeutic gene but also due to the inverse Grp78-activating effect of the combination of HSVtk with GCV, leading to stress. The combination of the transcriptional targeting and ligand-directed targeting of a gene therapy vector has proven to be effective in the suicide therapy of glioblastoma enhanced by the chemotherapy drug temozolomide (TMZ) [93]. This cytotoxic agent is responsible, in particular, for the activation of the unfolded protein response (UPR) stress pathway leading to the increased expression of endogenous Grp78 in tumors [94]. The co-administration of TMZ and RGD4C-AAVP increased the activity of the Grp78 promoter in the hybrid vector, providing enhanced expression of the delivered transgenes. Moreover, the known synergy between TMZ and HSVtk against human glioblastoma further improves the efficiency of this systemic phage-mediated cancer suicide gene therapy.

Despite promising results, initial studies of AAVP-RGD4C also revealed a reduced efficiency of transgene expression by cells transduced with this vector [74]. The study of the mechanisms of RGD4C-AAVP penetration into target cells and its subsequent intracellular transport showed that the internalization of the hybrid virus into mammalian cells occurs by clathrin-mediated endocytosis induced by the binding of the virus to integrin receptors [95]. At the same time, most of the hybrid viral particles, as a result of internalization, are trapped in late endosomes-lysosomes, where they are further degraded. The tumor extracellular matrix [96], the electrostatic repulsion between the negatively charged surfaces of bacteriophage particles and eukaryotic cells [97] and the high proteasome activity in cancer cells [54] also turned out to be factors reducing the efficiency of the bacteriophage-mediated transduction of target cancer cells. The discovery of these barriers to RGD4C-AAVP-mediated cancer gene therapy has led to the search for other ways to improve the hybrid viral vector [2]. In 2014, an increase in HSVtk/GCV-mediated cancer cell death was demonstrated by creating positively charged AAVP/cationic polymer complexes that help prevent electrostatic repulsion between the virus and the cell and promote the endosomal escape of viral particles [97]. The preliminary degradation of the ECM of rat glioblastoma, human melanoma and glioblastoma with collagenase and hyaluronidase facilitated the diffusion and internalization of the chimeric bacteriophage vector [96]. In this study, the phage-mediated expression of HSVtk and the subsequent treatment of targets with GCV caused a marked decrease in the viability of cancer cells by 33–63%, depending on the cell type, compared with the control. Combinations of RGD4C-AAVP with specific proteasome inhibitors and the organic substance genistein also increased the efficacy of cancer suicide gene therapy [54,55].

The improvement of AAVP by incorporating functional peptides into capsid proteins has been shown to be effective in cancer immunotherapy trials [76,98]. The use of octreotide, a synthetic analogue of somatostatin, as the targeting ligand of the AAVP vector, provided the highly selective delivery of tumor necrosis factor α (TNF-α) to pancreatic neuroendocrine tumor cells expressing surface somatostatin receptors [76]. In another recent study, the insertion of a histidine-rich functional H5WYG peptide into the pVIII AAVP capsid protein promoted the endosomal escape of hybrid viral particles after their internalization into target tumor cells [98]. The incorporation of H5WYG into the viral capsid contributed to an increase in osmotic swelling and the destabilization of the endosome containing RGD4C-H5WYG-AAVP, resulting in the release of viral particles into the cell’s cytoplasm. RGD4C-H5WYG-AAVP has been successfully used to deliver TNF-α to human chondrosarcoma cells [78]. This cytokine gene therapy resulted in the significant apoptosis of SW1353 cells both in vitro and in vivo, systemically administered in model mice. TNF-α is an inflammatory cytokine with antivascular and antitumor activities. The binding of this protein to its natural receptors, especially TNFR1, leads to the activation of the MAPK pathways and induces death signaling, eventually triggering apoptotic events and an immune response [99]. Previously, AAVP-RGD4C was also successfully used to deliver TNF-α to the tumor vasculature in human melanoma xenografts [77]. The AAVP-mediated systemic delivery of TNF-α led to the tumor-specific expression of this cytokine, which provides apoptosis in the vasculature and tumor necrosis. At the same time, the introduction of AAVP-RGD4C-TNF-α demonstrated the absence of systemic toxicity and transgene expression in cells of healthy tissues of mice with melanoma xenografts. This same strategy has been successfully used for the treatment of canine soft tissue sarcoma [100]. Even though dogs developed a sustained immune response against the injected chimeric viral particles, the treatment proved to be highly effective. Multiple injections of the gene therapy drug resulted in a tumor reduction of at least 85% of the original size and in the complete recovery of some experimental animals. At the same time, the expression of the transgene delivered by this bacteriophage-based vector was observed exclusively in tumor cells. In another study, AAVP-RGD4C-TNF-α caused the death of human glioblastoma cells when administered systemically to model mice [101].

The creation of transmorphic Phage/AAV (TPA) particles for the targeted delivery and expression of IL-12, IL-15 and TNF-α is a promising approach in the field of phage-mediated cancer therapy [80]. Unlike vectors based on full-length phage genomes, TPA particles have a more compact structure and, although they have the external characteristics of a native filamentous bacteriophage, contain only the transgenic AAV DNA cassette. The relatively modest size of these particles provides better transgene delivery due to the greater diffusion through the extracellular space and the improved intracellular transport. Moreover, the creators of TPA particles indicate that the compact structure of the particles contributes to their more successful avoidance of neutralization by immune cells during the systemic administration. TPA particles have demonstrated highly selective and effective cytokine therapy for several types of aggressive solid tumors: human glioblastoma, mouse melanoma and mouse colon adenocarcinoma. The high accuracy of delivery, the safety of systemic administration and the increased efficiency of interaction with cancer cells compared to AAVP characterize TPA particles as a new stage in the development of phage-based vectors.

Filamentous bacteriophages are not the only prokaryotic viruses that have found their way into immunogenic cancer therapy. In 2020, Y. J. Hwang and H. Myung constructed a T7 bacteriophage that provides targeted delivery of the mammalian granulocyte-macrophage colony-stimulating factor (GM-CSF) cytokine expression cassette to mouse melanoma cells [81]. This cytokine therapy resulted in the arrest of tumor growth and a significant reduction in tumor size in vitro and with the systemic administration of the phage in vivo. Animals injected with therapeutic phage showed increased survival compared to control mice. The authors of the work suggest that the expression of GM-CSF by transduced cancer cells leads to a local increase in the concentration of this cytokine in the tumor area, thereby attracting and activating cells of the immune system.

Finally, one of the most promising developments in phage-mediated cancer gene therapy to date is its combination with advanced CRISPR/Cas9 genome editing technology. In a recent study, RGD4C-H5WYG-AAVP provided the effective delivery of elements of the CRISPR/Cas9 system as the first step in targeted p53 gene replacement therapy for human lung cancer [79]. Lung adenocarcinoma cells are characterized by frequent mutations in the tumor suppressor p53 gene (TP53). Thus, the restoration of functional wild-type p53 should contribute to the suppression of target cancer cells growth and progression [102]. Cancer cells successfully transduced with the phage-based target vector turned out to be p53 knockout as a result of CRISPR/Cas9 expression [79]. The co-delivery of CRISPR/Cas9 targeting the mutant TP53 gene and a functional p53 protein gene by a bacteriophage-based vector is believed to have great potential for the treatment of various types of cancer. The integration of such a large transgenic cassette into the filamentous phage genome, however, can be difficult. This problem can be solved by using other bacteriophages with a larger genome capacity as cancer gene therapy vectors. As shown in 2019, the potential to create a hybrid prokaryotic-eukaryotic viral vector by combining T4 phage and AAV opens up the prospect of using this platform to deliver much more genetic and drug cargo than current vectors can provide [103]. The T7 phage is also currently being considered as a potential vector for efficient gene delivery to eukaryotic cells [104].

To date, bacteriophages have been used as gene therapy vectors for cancer in an increasing number of studies and clinical trials. Despite the fact that eukaryotic viruses have a much greater evolutionarily determined efficiency of mammalian cell transduction, which previously defined them as the most preferred sources of gene therapy vectors, their natural tropism to eukaryotic host cells makes their therapeutic application very difficult. The oncogenic potential of retroviral and lentiviral vectors, the highly time-limited expression of the target transgene and the high immunogenicity of adenovirus-based vectors slow down the development of new gene therapy approaches for cancer [105]. The creation of vectors based on recombinant AAV seems to be most effective, but its use is also constrained by its modest packaging capacity and the need to solve the problem of neutralizing antibodies and the selectivity of transduction during systemic administration [106]. At the same time, the unique features of the structure and biology of bacteriophages have led to the prospect of creating completely innovative methods of therapeutic gene delivery to target cancer cells.

## 5. Conclusions

In fact, the invention of phage display technology has turned bacteriophages into a powerful tool of molecular biomedicine. The easy modifiability and genetic plasticity of these viruses have made it possible to use engineered bacteriophages as therapeutic and diagnostic tools for various types of diseases, including cancer [107]. To date, bacteriophage-mediated cancer therapy has repeatedly proved its promise in a variety of preclinical trials. Apparently, in the near future, we will see even more studies using phage-based vectors as highly selective vehicles for therapeutic transgenes into target tumor cells. The engineered phage particles provide a safer and more accurate systemic delivery of therapeutic cargo to cancer cells compared to vectors based on eukaryotic viruses [108]. The search for ways to increase the effectiveness of cancer gene therapy using bacteriophages contributed to the development of new vector systems and transform particles that demonstrate extremely high levels of targeted gene expression in eukaryotic cells [80,98]. Such vehicles can provide improvements in experimental approaches to the treatment of oncological diseases based on the delivery of therapeutic genetic material. For example, combining these next-generation vectors and CRISPR/Cas9 genome editing technology represents a promising approach to cancer gene therapy [79]. The further optimization of this platform may lead to the creation of controlled nanoparticles that provide highly selective nuclease activity aimed at eliminating mutant oncogenes.

However, the further development of phage-mediated gene cancer therapy still requires the study of ways to create new vectors and improve existing vectors based on recombinant prokaryotic viruses. Although the chimeric AAVP vector is being used for targeted gene therapy for an increasing number of cancers, exploiting the unique strengths of other bacteriophages to create new hybrid viruses could raise the diversity of therapeutic approaches [103]. A future detailed study of the molecular aspects of the interaction between bacteriophages and eukaryotic cells, including cancer cells, will probably reveal new ways to improve phage-mediated transduction. In particular, the study of the mechanisms and factors underlying such a mysterious process as phage capsid deproteinization in eukaryotic cells seems to be essential for the further modernization of phage-based vectors. The mechanisms of phage gene expression in eukaryotic cells are also an important but poorly understood biological issue. Despite the existence of undeniable evidence of bacteriophage protein synthesis in mammalian cells, it is still not known for certain how cellular enzymes and transcription and translation factors carry out this process [59,109]. Although it has been established that the DNA of some phages may contain sequences homologous to cis-regulatory elements of eukaryotic genomes, the question of the existence of other prokaryotic gene expression methods remains open [60]. The search for and study of these mechanisms are necessary to understand the nature of phage–eukaryotic interaction and, as a result, to implement new biotechnological approaches using both native and engineered bacteriophages. The discovery of the ability of the bacteriophages M13 and T4 to actively induce a change in the expression profile of specific survival and signaling factors of tumor cells suggests that phage therapy can be used in cancer combination therapy [61]. More thorough studies of the molecular mechanisms of the interaction of bacteriophage particles with cancer cells contribute not only to the search for therapeutic agents but also to a deeper understanding of the cancer’s nature.

Cancer therapeutic platforms based on bacteriophages represent a prime example of innovative biotechnologies. The enormous plasticity of their modifiability, proven safety, growing efficacy and variety of their therapeutic applications are expanding the horizons of cancer gene therapy. Numerous successful and promising trials of phage-mediated tumor cell transduction suggest that the vast potential of these viruses for cancer gene therapy has yet to be fully exploited.

## Figures and Tables

**Figure 1 ijms-23-14245-f001:**
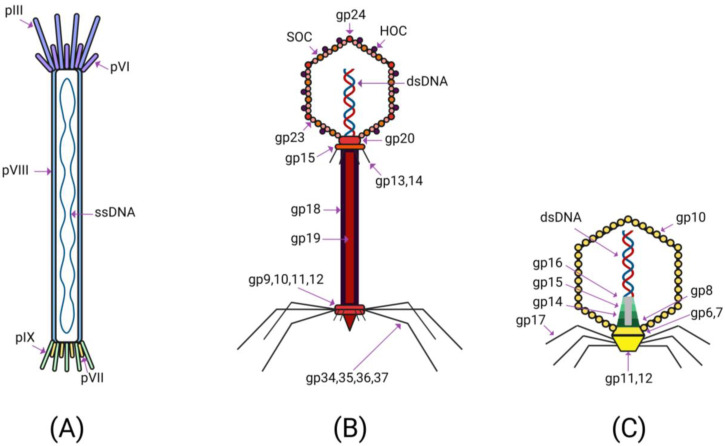
The structures of (**A**) M13 phage, (**B**) T4 phage and (**C**) T7 phage.

**Figure 2 ijms-23-14245-f002:**
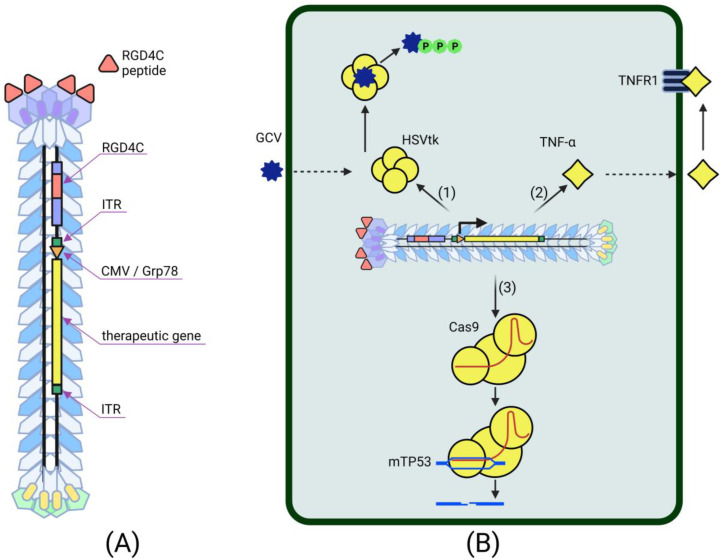
(**A**) Structure of RGD4C-targeted AAV/phage (AAVP) hybrid virus. The chimeric virus displays several copies of the RGD4C peptide on its pIII proteins to serve as a targeting ligand which specifically binds to β3 integrin receptors. The AAVP contains a therapeutic gene expression cassette flanked by AAV inverted terminal repeats (ITRs) and includes the CMV or Grp78 promoter; (**B**) Cancer gene therapy using the AAVP vector. (1) Suicide gene therapy mediated by the transformation of ganciclovir (GCV) to the toxic metabolite GCV-triphosphate under the influence of the herpes simplex virus thymidine kinase (HSVtk) expressed by AAVP in target cancer cells. (2) The delivery and expression of tumor necrosis factor α (TNF-α) by the AAVP in cancer cells leads to the release of this multifunctional protein and its binding to cell-surface receptors. The interaction of TNF-α with tumor necrosis factor receptor 1 (TNFR1) leads to the activation of various apoptotic pathways and cancer cell death, mediating cytokine gene therapy. (3) The goal of p53 gene replacement therapy is to disable the defective mutant tumor suppressor p53 gene (mTP53) and to simultaneously introduce the native functional p53 gene. Disabling the mTP53 can be mediated by combining the AAVP vector with clustered regularly interspaced short palindromic repeat/CRISPR-associated protein 9 (CRISPR/Cas9) technology, targeting the desired nucleotide sequence and generating DNA double-strand breaks.

**Table 1 ijms-23-14245-t001:** Advantages and limitations of phage-based vectors used in tumor-addressing cancer gene therapy.

Phage-Based Vector	Delivered Transgene	Advantages	Limitations	References
AAVP	HSVtk TNF-α CRISPR/Cas9	Easy modifiability and production of viral particles; Lack of non-targeted expression	Initially low cell transduction efficiency compared to eukaryotic virus-based vectors; Increased phage capsid length, as a result of the insertion of large transgenic cassettes, leads to limitations in packaging, cloning capacity and susceptibility to clearance by the reticuloendothelial system	[74,75] [76,77,78] [79]
TPA	IL-12, IL-15 TNF-α	Increased efficiency of target cell transduction due to reduced viral particle size; Extremely high yield of viral particles	A helper phage is required for the production of TPA particles	[80]
T7	GM-CSF	Large packaging capacity	Poor study as a vector for systemic delivery	[81,82]

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
