# Peer review of "Bacteriophage-Mediated Cancer Gene Therapy"

_ijms, 2022, doi:10.3390/ijms232214245_

Round 1
Reviewer 1 Report
The current review paper describes the potential of bacterial viruses, also known as bacteriophages, for development of gene delivery platforms for cancer treatment. After providing an overview of the biology and structure of bacteriophages, the authors discuss on the interaction between these bacterial viruses and eukaryotic cells and then reflect on the platforms developed thus far for phage-mediated gene therapy of cancer.
The current work is an interesting article. However, there are major concerns about the paper, which need to be addressed in detail. The manuscript suffers from not describing some important aspects of the phage-mediated gene delivery as well as not having proper referencing in some sections.
Major points
1. Phage display is one of the most important applications of bacteriophages with huge potential for translation into the clinic including development of cancer gene therapy platforms. It has been mentioned briefly. The authors need to explain phage display and its applications for cancer gene therapy in greater detail. It is suggested to be in Chapter 4 of the manuscript, not in Chapter 1.
2. The potential of bacteriophages for the development of cancer vaccines should be discussed in the manuscript. It is suggested to be detailed within the context of phage display.
3. The intracellular (cytoplasmic) fate of bacteriophages inside eukaryotic cells is an important aspect of phage-mediated gene therapy. However, it has been explained very briefly in Chapter 3 (lines 257-269). The authors need to address this important topic extensively and discuss all the intracellular barriers for phage-mediated gene expression. This should also be described within the evolutionary relationship between bacteriophages and eukaryotic cells and how phages lack the molecular machinery developed by eukaryote cell-specific viruses. Also, it should be delineated how phages can be modified, inspired by eukaryotic viruses, to overcome these intracellular barriers.
4. The use of NLS in engineering phages to improve phage-mediated gene delivery should be discussed more, describing how it can improve the expression of phage-encoded genes inside eukaryotic cells.
5. The manuscript contains many grammatical as well as punctuation errors, which need to be checked thoroughly to improve English language.
Minor points
Line 70: Replace “creatures” with “biological entities”.
Line 144-145: Add “translation” as an important step of the phage life cycle (after replication and transcription).
Line 160: Add the seminal paper by George Smith, in which he introduced “phage display” (doi: 10.1126/science.4001944), as the reference.
Line 168: Add a reference for the higher titer of lysogenic (like filamentous) phages compared to lytic phages.
Line 168: Remove “freely”. There are many restrictions for display of peptides on the surface of bacteriophages (depending on the type of the phage, surface protein, etc).
Line 170: Use the phrase “phenotype-genotype link” for explanation of number 3.
Line 173: The displayed peptide is not encoded by a gene. Replace “gene” with “encoding sequence”.
Line 196: The sentence needs a reference.
Line 199: It seems that the first evidence on the interaction of phages with mammalian cells dates back to 1940s and the work of Bloch. The original paper is in German, but cited in these papers (doi: 10.1111/j.1365-2672.2004.02422.x and doi: 10.1517/17425247.2014.927437).
Line 203: “affect” is not a clear word. It has been used in different place of the manuscript. It is suggested to be replaced with “interaction”.
Line 211-222: In this paragraph, it is highly needed to cite the first study conducted by Larocca et al. in which the ability of bacteriophages for gene delivery to eukaryotic cells was shown.
Lines 302-304: How do you conclude that bacteriophages play roles in the course of cancer? The reference 61 only concludes that bacteriophages (M13 and T4) interact with and influence gene expression in LNCaP cells.
Line 389: The phrase “tumor vasculature cells” does not seem to be correct.
Round 1
Response to Reviewer 1 Comments
We thank reviewer for a thorough revision of our manuscript. Please find below our response to reviewer’s comments.
Major Point 1:
Phage display is one of the most important applications of bacteriophages with huge potential for translation into the clinic including development of cancer gene therapy platforms. It has been mentioned briefly. The authors need to explain phage display and its applications for cancer gene therapy in greater detail. It is suggested to be in Chapter 4 of the manuscript, not in Chapter 1.
Response 1:
Corrected. We have added more information to the description of the phage display and its applications for cancer gene therapy and moved it to Chapter 4.
Major Point 2:
The potential of bacteriophages for the development of cancer vaccines should be discussed in the manuscript. It is suggested to be detailed within the context of phage display.
Response 2:
Corrected. A detailed description of the principles of phage-based cancer vaccine development is added.
Major Point 3:
The intracellular (cytoplasmic) fate of bacteriophages inside eukaryotic cells is an important aspect of phage-mediated gene therapy. However, it has been explained very briefly in Chapter 3 (lines 257-269). The authors need to address this important topic extensively and discuss all the intracellular barriers for phage-mediated gene expression. This should also be described within the evolutionary relationship between bacteriophages and eukaryotic cells and how phages lack the molecular machinery developed by eukaryote cell-specific viruses. Also, it should be delineated how phages can be modified, inspired by eukaryotic viruses, to overcome these intracellular barriers.
Response 3:
Corrected. We discussed in more detail the topic of intracellular fate of phages, existing and currently known intracellular barriers to phage-mediated gene expression.
Major Point 4:
The use of NLS in engineering phages to improve phage-mediated gene delivery should be discussed more, describing how it can improve the expression of phage-encoded genes inside eukaryotic cells.
Response 4:
Corrected. We added an evaluation of the potential of using NLS in phage engineering to improve phage-mediated gene delivery.
Major Point 5:
The manuscript contains many grammatical as well as punctuation errors, which need to be checked thoroughly to improve English language.
Response 5:
Corrected. We conducted an additional analysis of the text for grammatical and punctuation errors.
Minor points
We also took all the minor points into account and made adjustments accordingly. We are grateful to the reviewer for careful reviewing and close attention to our article.
Corrected text is displayed in red color.
We highlighted (red highlighted text) all changes made when revising the manuscript to make it easier for the Editors to give a prompt decision on manuscript.

Reviewer 2 Report
This is an excellent, well-written review paper on bacteriophages used in cancer gene therapy. It appropriately describes the field and is strongly and thoroughly cited. This is an important contribution to biotechnology and gene therapy.
Round 1
Response to Reviewer 2 Comments
We thank reviewer for a thorough revision of our manuscript. Please find below our response to reviewer’s comments.
Point 1:
This is an excellent, well-written review paper on bacteriophages used in cancer gene therapy. It appropriately describes the field and is strongly and thoroughly cited. This is an important contribution to biotechnology and gene therapy.
Response 1:
We thank the reviewer for his thoughtful, thorough and benevolent review.

Reviewer 3 Report
Gleb Petrov et al reviewed the key result of research into aspects of phage-eukaryotic cell interaction and phage-based vector to cancers.
It is very interesting review of bacteriophages-based gene transfer, but I point out some problems.
1. Can you show the advantages of bacteriophages-based gene transfer compared to previous virus or non-viral vector in abstract?
2. Can you add sub-headings more in your review?
3. Can you table the advantages and disadvantages of various gene transfer using bacteriophages?
4. Can you compare the function of bacteriophages-based gene transfer, lentivirus, adenovirus, AAV, retrovirus, and HSV-1 etc.
Round 1
Response to Reviewer 3 Comments
We thank reviewer for a thorough revision of our manuscript. Please find below our response to reviewer’s comments.
Point 1:
Can you show the advantages of bacteriophages-based gene transfer compared to previous virus or non-viral vector in abstract?
Response 1:
Corrected. We have added the relevant information to the article abstract. Since the abstract has strict space limitations, we also had to remove some sentences of the abstract.
Point 2:
Can you add sub-headings more in your review?
Response 2:
Corrected. We added more subheadings to the article structure.
Point 3:
Can you table the advantages and disadvantages of various gene transfer using bacteriophages?
Response 3:
Corrected. We have added a table with the advantages and disadvantages of phage-based vectors used in cancer gene therapy and mentioned in the review.
Point 4:
Can you compare the function of bacteriophages-based gene transfer, lentivirus, adenovirus, AAV, retrovirus, and HSV-1 etc.
Response 4:
Corrected. We have added relevant information to the end of the "Bacteriophage-mediated Cancer Gene Therapy " chapter.
Corrected text is displayed in red color.
We highlighted (red highlighted text) all changes made when revising the manuscript to make it easier for the Editors to give a prompt decision on manuscript.

Round 2
Reviewer 1 Report
The authors have addressed all the points properly.